

# Surveillance of tuberculosis and treatment outcomes following screening and therapy interventions among marriage-migrants and labor-migrants from high TB endemic countries in Taiwan

Mei-Mei Kuan

Chief Secretary Office, Taiwan Centers for Disease Control, Ministry of Health and Welfare, Taipei, Taiwan

## ABSTRACT

**Background:** Tuberculosis (TB) among migrants from high-risk countries and underling interventions were concerned for disease control. This study aimed to assess the TB trends among marriage-migrants with the 1–2-round vs. labor-migrants with the four-round TB screenings in the period of the first four post-entry years; pre-entry screenings by an initial chest X-ray (CXR) were conducted during 2012–2015, and a friendly treatment policy was introduced in 2014.

**Methods:** TB data of migrants during 2012–2015 were obtained from the National TB Registry Database and analyzed. The incidences, clinical characteristics, and treatment outcomes were assessed to explore the impact of underlying interventions.

**Results:** During post-entry 0–4 years, the TB incidence rates among marriage-migrants ranged 11–90 per 100,000 person-years, with 60.8% bacteria-positive and 28.2% smear-positive cases. Whereas among labor migrants, the incidence rates ranged 67–120 per 100,000 person-years, with 43.6% bacteria-positive and 13.7% smear-positive cases. All migrants originated from Southeast Asia following pre-entry health screening in 2012–2015. The TB cases among marriage-migrants were with a higher proportion of sputum-smear-positivity (SS+) (OR: 4.82, 95% CI [3.7–6.34]) and CXR cavitation (OR: 2.90, 95% CI [2.10–4.01]). Marriage-migrants with TB had treatment completion rate of >90%, which was above the WHO target. For labor-migrants with TB, when compared the period of post- vs. pre-implementation of the friendly therapy policy that eliminated compulsory repatriation, the overall treatment completion rate of those who stayed in Taiwan improved by 30.9% (95% CI [24.3–37.6]) vs. 6.7% (95% CI [3.8–9.7]), which exceeded a 4.88-fold (95% CI: 3.83–6.22) improvement. Additionally, the treatment initiation rate within 30 days of diagnosis for SS- TB and B- TB cases during post- vs. pre-implementation of the therapy policy was increased, that is, 77.1% vs. 70.9% (OR: 1.38, 95% CI [1.12–1.70]) and 78% vs. 77% (OR: 1.64, 95% CI [1.38–1.95]).

**Conclusion:** Multiple CXR screenings could identify more TB cases with sputum-smear-negativity (SS-) TB at the early-stage, introducing latent tuberculosis infection (LTBI) screening might save underlying efforts. For those labor-migrants with TB

Corresponding author
Mei-Mei Kuan, kuan@cdc.gov.tw

who stayed in the receiving country, the friendly TB therapy policy not only significantly improved the treatment completion but also the early treatment initiation.

# INTRODUCTION

Taiwan, with a population of approximately 23.4 million, is a country with moderate TB incidence ranged 46–53 per 100,000 population in 2012–2015 (*Taiwan Centers for Disease Control, 2018*) and has gradually decreased over time. Nevertheless, TB disproportionately affects the foreign-born population from TB high-risk countries, with an incidence in this population that is several times higher than that among the Taiwanese-born population (*Global Tuberculosis Report, 2018*; *The World Bank, 2018*; *Taiwan Centers for Disease Control, 2018*). The reasons for the TB burden in the migrant population are likely to be the reactivation of remotely acquired LTBI following migration from high TB burden countries to lower TB burden countries (*Global Tuberculosis Report, 2018*; *The World Bank, 2018*; *Taiwan Centers for Disease Control, 2018*). TB control measures for the foreign-born population from high endemic countries in Taiwan occur within the scopes of the TB medical surveillance program. All newly arriving residents and temporary residents with high risk undergo an immigration medical examination before arrival. This examination consists of a physical examination which includes chest X-ray (CXR), syphilis test, HIV serology and other routine tests. If there is a radiographic evidence of TB, three sputum smears and mycobacterial cultures are examined (*Taiwan Centers for Disease Control, 2018*). Following the pre-entry screening, mandatory TB screening of either four rounds during the first 3 years (*Kuan, 2018*) or 1–2 rounds during the first 4 years (*Kuan, Yang & Wu, 2014*) are required post-entry for labor migrants or marriage migrants, respectively, who come from Southeast Asian (including Indonesia, Vietnam, Philippines, and Thailand) or China. Applicants with active TB are required to complete treatment before entering Taiwan. Before 2013, migrant workers with TB were repatriated; since 2014, these individuals are allowed to stay in Taiwan for treatment except multiple drug-resistant TB (MDR-TB) patients. A study showed that TB case with smear-positive sputum was more infectious than those with smear-negative sputum (*Kuan, 2018*). Our previous studies revealed that a 30.2% (*Kuan, Yang & Wu, 2014*) and 14.3% (*Kuan, 2018*) smear-positive sputum among marriage migrants vs. labor migrants in Taiwan. In term of TB control targeting high TB incidence migrants, it should be taken into consideration along with screening the differences in pathogen exposure due to their origins, the ethnic socio-economic disparities and the experience of migration itself (*Hayward et al., 2018*); moreover, to overcome the cultural and structural barriers to accessing healthcare (*Hayward et al., 2018*) is also critical. Thus, a friendly TB therapy policy may prompt access to treatment for a migrant in the new residing country. This study retrospectively analyzed a cohort from 2012–2015 aimed to analyze the impact

on TB incidence and clinical characteristics following the enhanced screening interventions among these high-risk migrants. Also, the treatment outcomes following the concurrent TB intervention programs as well as the effectiveness of introducing a friendly TB therapy policy in 2014 targeting labor migrants was assessed.

## MATERIALS AND METHODS

### Data Collection

TB data on migrants were retrieved from the following sources similar to previous studies (*Kuan, 2018*; *Kuan, Yang & Wu, 2014*): (1) the national TB registry database, which was updated regularly by clinicians and local health divisions; (2) The populations of marriage migrants and labor migrants were obtained from the official publications of the Ministry of the Interior (*Ministry of Interior Taiwan, 2016*).

### Definition of migrants with TB

Two strategies of post-entry TB screenings were targeted towards foreign migrants from highly endemic Southeast Asian countries (including Vietnam, Philippines, Thailand, and Malaysia) during the first 4 post-entry years following pre-entry screenings in Taiwan: (1) For labor migrants (foreign workers), 4 screenings at 0–3 days as well as at 6, 12, 18 and 30 months are conducted (*Kuan, 2018*). (2) For marriage migrants (foreign spouses), one to two mandatory TB screenings are conducted (*Kuan, Yang & Wu, 2014*). The definition of marriage migrants with TB (*Kuan, Yang & Wu, 2014*) or labor migrants with TB (*Kuan, 2018*) was described as previously (*Kuan, 2018*; *Kuan, Yang & Wu, 2014*). Treatment completion, according to the WHO guidelines, is defined as accomplished by a standard regimen, that is, a 6–9-month regimen, or complete treatment by a longer regimen. The TB treatment completion rate was obtained as the number of TB patients who achieved treatment completion by the standard or longer therapy regimen divided by the number of TB patients staying in Taiwan.

### Statistics and analysis

The TB incidence (per 100,000 population) was calculated as the number of newly detected labor migrants with TB during the study period (2012–2015) divided by the population per year. The yearly fluctuating trends in the TB incidence rates among labor migrants ($X_1$) and marriage migrants ($X_2$) vs. the WHO estimated TB incidence rates (*Taiwan Centers for Disease Control, 2018*) ($Y$), as well as the TB incidences ($X_1$) of the labor migrants vs. the marriage migrants ($X_2$), were detected with linear regression analysis. Furthermore, several odds ratio tests were performed: the studied TB case population was divided into two exposure categories: marriage migrants vs. labor migrants or post-implementation (2014–2015) vs. pre-implementation (2012–2013) of the friendly therapy policy. Categorical variables were subjected to binary analysis using a 2 × 2 table to compute odds ratios (ORs) and 95% confidence intervals (CIs) by using R Programming or OpenEpi (*Dean, Sullivan & Soe, 2006*) for assessment of the difference between paired binary variable determinants; the control (i.e., reference) for each variable was the total number in each stratified category, excluding the detected items (i.e., independent

variables). The TB treatment outcome was estimated by calculating the transferred-out rate and the treatment completion rate. To assess the impact of implementing the friendly therapy policy with no compulsory repatriation, the post-implementation (2014–2015) vs. pre-implementation (2012–2013) TB treatment completion rate, the percentage of sputum smear positivity of the transferred out cases vs. the cases staying in Taiwan for therapy, and the initiation of treatment within 30 days of diagnosis, that is, the delay (in days) of treatment initiation (standard or longer regimen) following diagnosis among labor migrants during post vs. pre-implementation of the therapy policy, were estimated by $t$-tests (2-sides) or Chi-squared tests.

### Ethics statement

This study was approved by the Institutional Review Board of the Taiwanese CDC under identification number TwCDCIRB106135.

## RESULTS

### TB incidence trends among migrants and TB screening impact

The TB surveillance strategy in Taiwan implemented a mandatory screening post-entry for immigrants at risk. This is either an 1–2 rounds or four rounds of screenings targeted to marriage migrants or labor migrants who come from high-burden countries, during the first 0–4 years or the first 0–3 years, respectively, following an initial pre-entry screening with CXR detection. For migrants from the same area of origin (Indonesia, Vietnam, Thailand, and the Philippines), the TB incidence rates of labor migrants were higher than those of marriage migrants, $P < 0.001$ (Table 1). Briefly, the TB incidence ranged 67.71–143.78 (per 100,000) among labor migrants, 11.3–99.74 among marriage migrants from Southeast Asia, and 11.4–15.66 among those from China during their first post-entry 0–4 years in 2012–2015 (Table 1). Relationships among the fluctuation trends during the first 0–4 years of the post-entry period were detected. The TB incidence rates of the labor migrants ($X_1$) or of the marriage migrants ($X_2$) were positively correlated with those of WHO estimated (Y) for their countries of origin (*The World Bank, 2018*) based on yearly surveillance by the Pearson's correlation or liner association in the manner of significant ($n = 16$, Pearson's correlation $R_{X1:Y}$: 0.74, linear association $R^2_{X1:Y}$: 0.55, $P < 0.0001$) or of less significant ($n = 14$, Pearson's correlation $R_{X2:Y}$: 0.65, linear association $R^2_{X2:Y}$: 0.42, $P = 0.0125$). Furthermore, the TB incidence rates of the labor migrants ($X_1$) vs. those of the marriage migrants ($X_2$) identified were higher, but the association between these fluctuation trends appeared insignificant (i.e., $n = 14$, Pearson's correlation $R$:0.248; linear association: $R^2_{X2:X1}$: 0.061, $P = 0.393$).

### Comparison of TB screening impact on clinical characteristics among TB cases

TB cases identified among marriage migrants were similar to those identified among labor migrants in terms of nationality, but the marriage migrants were more likely to be women (OR: 25.37, 95% CI [13.47–47.81]), were more likely to be older than 45 years old (OR: 3.89, 95% CI [2.61–5.78]) and had more severe infections (Table 2) at diagnosis.

**Table 1 Tuberculosis notification and treatment outcomes among labor and marriage migrants.**

| | | Labor migrants | | | | | Marriage migrants | | | | Chi-sq[g] | |
|---|---|---|---|---|---|---|---|---|---|---|---|---|
| Year | | 2012 | 2013 | 2014 | 2015 | Sum | 2012 | 2013 | 2014 | 2015 | Sum | p-value |
| | Notification TB cases | 472 | 581 | 646 | 624 | 2,323 | 109 | 92 | 95 | 84 | 380 | |
| Age | ≤24 | 101 | 102 | 129 | 120 | 452 | 6 | 9 | 4 | 4 | 23 | <0.001 |
| | 25–44 | 363 | 464 | 489 | 483 | 1,799 | 91 | 72 | 80 | 72 | 315 | |
| | ≤45 | 8 | 15 | 28 | 21 | 72 | 12 | 11 | 11 | 8 | 42 | |
| Sex | Female | 295 | 352 | 361 | 370 | 1,378 | 107 | 92 | 89 | 82 | 370 | <0.001 |
| | Male | 177 | 229 | 285 | 254 | 945 | 2 | 0 | 6 | 2 | 10 | |
| Country | Indonesia | 221 | 286 | 322 | 296 | 1,125 | 13 | 6 | 7 | 9 | 35 | <0.001 |
| | Vietnam | 77 | 98 | 102 | 123 | 400 | 44 | 44 | 34 | 30 | 152 | |
| | Philippines | 109 | 128 | 152 | 162 | 551 | 4 | 4 | 8 | 0 | 16 | |
| | Thailand | 65 | 69 | 68 | 43 | 245 | 0 | 2 | 3 | 1 | 6 | |
| | China | | | | | 0 | 48 | 36 | 42 | 44 | 170 | |
| Population | Indonesia | 191,127 | 213,234 | 229,491 | 236,526 | 870,378 | 27,684 | 27,943 | 28,287 | 28,699 | 112,613 | <0.001 |
| | Vietnam | 100,050 | 125,162 | 150,632 | 169,981 | 545,825 | 87,357 | 89,042 | 91,004 | 93,441 | 360,844 | |
| | Philippines | 86,786 | 89,024 | 111,533 | 123,058 | 410,401 | 7,465 | 7,707 | 8,021 | 8,326 | 31,519 | |
| | Thailand | 67,611 | 61,709 | 59,933 | 58,372 | 247,625 | 8,336 | 8,375 | 8,467 | 8,525 | 33,703 | |
| | China | | | | | 0 | 306,514 | 315,905 | 323,358 | 330,069 | 1,275,846 | |
| TB incidence | Indonesia | 115.63 | 134.13 | 140.31 | 125.14 | | 46.96 | 21.47 | 24.75 | 31.36 | | |
| | Vietnam | 76.96 | 78.3 | 67.71 | 72.36 | | 50.37 | 49.41 | 37.36 | 32.11 | | |
| | Philippines | 125.6 | 143.78 | 136.28 | 131.65 | | 53.58 | 51.9 | 99.74 | 0 | | |
| | Thailand | 96.14 | 111.82 | 113.46 | 73.67 | | 0 | 23.88 | 35.43 | 11.73 | | |
| | China | | | | | | 15.66 | 11.4 | 12.99 | 13.33 | | |
| Clinical characteristics | B+[a] | 205 | 260 | 271 | 276 | | 67 | 59 | 64 | 41 | | |
| | B–[b] | 234 | 290 | 349 | 325 | | 40 | 31 | 29 | 39 | | |
| | SS+[c] | 77 | 81 | 75 | 85 | | 31 | 32 | 31 | 23 | | |
| | MDR-TB | 1 | 6 | 2 | 3 | | 0 | 0 | 0 | 0 | | |
| TB treatment | DOST | 297 | 398 | 457 | 484 | | 101 | 92 | 92 | 80 | | |
| | Died | 1 | 2 | 2 | 3 | | 0 | 1 | 1 | 0 | | |
| | Transferred out | 426 | 526 | 457 | 380 | | 9 | 2 | 3 | 2 | | |
| | Transferred-out rate | 0.903 | 0.905 | 0.707 | 0.609 | 0.21 (0.16–0.26)[h] | 0.083 | 0.022 | 0.032 | 0.024 | | |
| | Lost to follow up | 8 | 9 | 11 | 15 | 1.27 (0.7–2.4)[h] | 0 | 1 | 1 | 1 | | |
| | Staying in Taiwan (ST)[d] | 46 | 55 | 189 | 244 | | 100 | 90 | 92 | 82 | | |
| | Treat. completion (TC)[e] | 46 | 48 | 178 | 233 | 4.88 (2.8–6.2)[h] | 99 | 89 | 90 | 80 | | |
| | TC = 6–9 months[e] | 42 | 48 | 174 | 232 | | 99 | 89 | 88 | 79 | | |
| | TC > 6–9 months[e] | 0 | 0 | 4 | 1 | | 0 | 0 | 2 | 1 | | |
| | STT-completion rate[f] | 0.913 | 0.873 | 0.92 | 0.95 | | 0.96 | 0.989 | 0.96 | 0.96 | | |

**Notes:**
[a] B+: bacterial positivity, with positivity among 3 sputum smears or sputum cultures.
[b] B–: bacterial negativity, with no positivity among 3 sputum smears or sputum cultures.
[c] SS+: positivity among 3 sputum smears.
[d] Staying in Taiwan cases = all TB cases—transferred-out cases, including compulsorily repatriated cases.
[e] Treatment completion using 6–9-month regimens, that is, a standard regimen or longer regimen.
[f] The treatment completion rate of cases staying in Taiwan = cases staying in Taiwan with treatment completion by a standard or longer therapy regimen/cases staying in Taiwan.
[g] Pearson's Chi-squared test.
[h] Odds ratio of TB cases during 2014–2015 vs. 2012–2013.
**Table 2 The impact of TB screening on clinical characteristics among migrants with TB.**

|  | Marriage migrant | Labor migrant | Odds Ratio |
|---|---|---|---|
| Total | N = 380 | N = 2,323 | |
| Sex | | | |
| Female | 370 | 1,378 | 25.37 (13.47–47.81)* |
| Age at diagnosis, years | | | |
| ≥45 | 42 | 72 | 3.89 (2.61–5.78)* |
| 44≤ | 338 | 2,251 | 0.26 (0.17–0.38)* |
| CXR | | | |
| Normal | 40 | 238 | 1.03 (0.72–1.47) |
| Abnormal without cavitation | 270 | 1,922 | 0.51 (0.40–0.66)* |
| Abnormal with cavitation | 60 | 141 | 2.90 (2.10–4.01)* |
| No record | 10 | 260 | |
| Sputum smear | | | |
| SS− | 246 (64.7%) | 1,834 (78.9%) | 0.49 (0.39–0.62)* |
| SS+ | 107 (28.2%) | 318 (13.7%) | 4.82 (3.7–6.34)* |
| No record | 27 | 171 | |
| Sputum culture | | | |
| SC+ | 222 | 990 | 1.89 (1.52–2.39)* |
| SC− | 148 | 1,222 | 0.57 (0.46–0.71)* |
| No record | 10 | 824 | |
| Bacterial status | | | |
| B+ | 231 (60.8%) | 1,012 (43.6%) | 2.01 (1.61–2.51)* |
| B− | 149 (39.2%) | 1,311 (56.4%) | 0.50 (0.40–0.62)* |

Notes:
* Odds ratio: significant.
SS, sputum smear; SC, sputum culture.

The effect of TB screenings on clinical characteristics among migrants with TB after post-entry screenings were assessed and revealed 60.8% bacteria-positive and 58.4% smear-positive cases among marriage migrants; 43.6% bacteria-positive and 42.6% smear-positive cases among labor migrants (Table 1). The marriage migrants with 1–2 rounds vs. labor migrants with four rounds post-entry screenings had higher rates of sputum smear positivity (SS+) (OR: 4.82, 95% CI [3.7–6.34]), higher rates of CXR cavitation (OR: 2.90, 95% CI [2.10–4.01]) and higher rates of B+ (OR: 2.1, 95% CI [1.61–2.51]) (Table 2).

## Anti-TB treatment outcomes: post- vs. pre-implementation of the friendly therapy policy

The overall treatment completion rate was >90% among both the labor and marriage migrants who stayed in Taiwan and accepted TB treatment (Table 1), which was above the WHO target of >85% (*Dean, Sullivan & Soe, 2006*). Comparing outcome during post- (2014–2015) vs. pre- (2012–2013) implementation of the therapy policy with no compulsory repatriation for labor migrants, the treatment completion rate was increased by 4.88-fold (95% CI [3.83–6.22]), that is, 30.9% (95% CI [24.3–37.6]) vs. 6.7% (95% CI

[3.8–9.7]) (Table 1). Higher completion rates for SS− TB cases OR: 7.45 (95% CI [5.44–10.2]) or B− TB cases OR: 6.87 (95% CI [4.21–11.2]); SS+ TB cases, OR: 13.43 (95% CI [4.02–44.79]) (Table 3), or B+TB cases OR: 5.83 (95% CI [3.62–9.42]) were also observed.

After 2014 (post-) vs. before 2013 (pre) implementing friendly therapy being less in transfer out TB cases of OR: 0.21 (95% CI [0.16–0.26]) (Table 1) which was detailed by less transfer out cases with both SS- TB cases of OR: 0.61 (95% CI [0.38–0.97]) or B− TB cases of OR: 0.58 (95% CI [0.49–0.69]) and SS+ TB cases of OR: 0.44 (95% CI [0.27–0.71]) or B+ TB cases of OR: 0.66 (95% CI [0.56–0.79]) (Table 3) as well as being no significant change in lost-to follow up TB cases of OR: 1.27 (95% CI [0.69–2.40]) among labor migrants were exhibited (Table 1).

In terms of overall TB treatment initiation within 30 days of diagnosis, in both migrants with SS+ TB (96.7% (188/194) vs. 91.2% (290/318)) and SS− TB (81.8% (301/368) vs. 74.3% (1,364/1,834)), the marriage migrants were more compliant than labor migrants. Moreover, comparing the overall initiation rate of TB treatment among labor migrants with TB showed a significant increase of 73.7% vs. 68.2% (OR: 1.31, 95% CI [1.09–1.57]) during the post- vs. pre-implementation of the therapy policy. Furthermore, during the post- vs. pre-implementation of the therapy policy, the TB treatment initiation rate among labor migrants showed a significant increase of 77% vs. 71% (OR: 1.38, 95% CI [1.12–1.70]) with SS− TB and a non-significant change of 89.4% vs. 93.00% (OR: 0.63, 95% CI [0.29–0.39]) with SS+ TB. Meanwhile, a significant increasing of 78% vs. 77% (OR:1.64, 95% CI [1.38–1.95]) with B− TB and a non-significant change of 83% vs. 85% (OR: 0.84, 95% CI [0.70–1.01]) with B+ TB was observed (Table 3).

## DISCUSSION

Multiple initial screenings with CXR could identify more SS− TB cases at an early stage with low infectivity and result in a higher incidence. For migrants who come from high TB burden regions, a pre-entry initial screening with a CXR is required and follow by mandatory 1–2 rounds of post-entry screening for marriage migrants or four rounds for labor migrants in Taiwan, respectively. This initiative resulted in a higher incidence with more SS− TB among labor migrants and a lower incidence but more SS+ TB or more CXR cavitation TB cases among marriage-migrants (Tables 1 and 2). Moreover, the higher annual TB incidence rates in labor migrants were significantly ($R^2$: 0.55, $P < 0.0001$) associated with the WHO estimated TB incidence rates (*Taiwan Centers for Disease Control, 2018*) of their original countries and there were significantly fewer severe cases, that is, fewer SS+ TB cases or fewer CXR cavitation TB cases, among the labor migrants. On the other hand, relatively lower annual TB incidence rates among marriage migrants were less significantly associated with the WHO-estimated TB incidence rates (*Taiwan Centers for Disease Control, 2018*) of their original countries ($R^2 = 0.42$, $P = 0.0125$), and there were significantly more severe cases, that is, more SS+ TB cases (OR: 4.82), or more CXR abnormal with cavitation TB (OR: 2.90) (Table 2). Thus, these results indicated that the greater number (four rounds) of mandatory post-entry screenings for labor migrants vs. the 1–2 rounds for marriage migrants could screen out more TB cases;

**Table 3 Tuberculosis treatment outcomes of labor migrants with TB during post- vs. pre- friendly therapy policy.**

| Anti-TB treatment | TB cases | Pre-FT $N = 1{,}053$ | Post-FT $N = 1{,}270$ | Odds ratio |
|---|---|---|---|---|
| Outcome by nationality | Indonesia | $n' = 507$ | $n' = 618$ | |
| | Transf. out | 450 | 394 | 0.22 (0.16–0.31)* |
| | Longer reg. | 0 | 4 | |
| | Stand. Reg. | 49 | 206 | 4.81 (3.43–6.75)* |
| | No record | 8 | 14 | |
| | Vietnam | $n' = 175$ | $n' = 225$ | |
| | Transf. out | 158 | 153 | 0.23 (0.13–0.40)* |
| | Longer reg. | 0 | 0 | |
| | Stand. Reg. | 15 | 67 | 4.52 (2.48–8.25)* |
| | No record | 98 | 5 | |
| | Philippines | $n' = 237$ | $n' = 314$ | |
| | Transf. out | 212 | 206 | 0.22 (0.14–0.36)* |
| | Longer reg. | 0 | 1 | |
| | Stand. Reg. | 24 | 106 | 4.52 (2.79–7.33)* |
| | No record | 1 | 1 | |
| | Thailand | $n' = 134$ | $n' = 111$ | |
| | Transf. out | 132 | 84 | 0.05 (0.01–0.20)* |
| | Longer reg. | 0 | 0 | |
| | Stand Reg. | 2 | 26 | 19.88 (4.6–85.95)* |
| | No record | 0 | 1 | |
| Outcome of DOST | SS-/DOST | $n = 537$ | $n = 784$ | 1.60 (1.3–1.96)* |
| | Completion | 50 | 334 | 7.45 (5.44–10.2)* |
| | Refuse | 4 | 5 | |
| | Transfer out | 250 | 225 | 0.61 (0.38–0.97)* |
| | Lost to follow up | 14 | 22 | |
| | Side effect | 5 | 5 | |
| | Not bac | 33 | 16 | |
| | other | 170 | 175 | |
| | SS+/DOST | $n = 124$ | $n = 117$ | 0.75 (0.45–1.25) |
| | Completion | 3 | 33 | 13.43 (4.02–44.79)* |
| | Transfer out | 62 | 45 | 0.44 (0.27–0.71)* |
| | Lost to follow up | 3 | 4 | |
| | Side effect | 3 | 0 | |
| | Not bac | 1 | 0 | |
| | other | 49 | 35 | |
| | B+/DOST | $n = 332$ | $n = 382$ | 0.93 (0.78–1.11) |
| | Refuse | 5 | 1 | |
| | Complete | 20 | 129 | 5.83 (3.62–9.42)* |
| | Side effect | 3 | 3 | |
| | Lost to follow up | 11 | 8 | |
| | Transfer out | 438 | 408 | 0.66 (0.56–0.79)* |

| Table 3 (continued) | | | | |
| --- | --- | --- | --- | --- |
| **Anti-TB treatment** | **TB cases** | **Pre-FT** $N = 1{,}053$ | **Post-FT** $N = 1{,}270$ | **Odds ratio** |
| | B-/DOST | $n = 346$ | $n = 534$ | 1.48 (1.25–1.76) |
| | Refuse | 8 | 4 | |
| | Complete | 33 | 240 | 6.87 (4.21–11.2)[*] |
| | Side effect | 5 | 2 | |
| | Lost to follow up | 6 | 18 | |
| | Transfer out | 458 | 392 | 0.58 (0.49–0.69)[*] |
| Treatment Initiation day of diagnosis | SS− | $n = 805$ | $n = 1{,}029$ | |
| | <30 day | 571 | 793 | 1.38 (1.12–1.70)[*] |
| | >30 day | 145 | 157 | 0.82 (0.64–1.05) |
| | No record | 89 | 79 | |
| | <$n$ 30 day rate[**] | 0.71 | 0.77 | |
| | SS+ | $n = 158$ | $n = 160$ | |
| | <30 day | 147 | 143 | 0.63 (0.29–1.39) |
| | >30 day | 2 | 3 | 2.99 (0.50–18.08) |
| | No record | 9 | 14 | |
| | <30 day rate | 0.93 | 0.89 | |
| | B− | $n = 399$ | $n = 658$ | |
| | <30 day | 307 | 512 | 1.64 (1.38–1.95)[*] |
| | >30 day | 92 | 146 | |
| | <30 day rate[**] | 0.77 | 0.78 | |
| | B+ | $n = 347$ | $n = 379$ | |
| | <30 day | 296 | 313 | 0.84 (0.70–1.01) |
| | >30 day | 51 | 66 | |
| | <30 day rate | 0.85 | 0.83 | |

**Notes:**

[*] Comparison of treatment outcomes post- vs. pre- implementation of friendly therapy, odds ratio test: significant; either $N$ or $n'$: calculate reference for odds ratio.

[**] The proportions of cases of initiate treatment between post-FT and pre-FT was significantly different which was approved by a Fisher's exact test, $P = 0.0296$ ($P < 0.05$) that is, to reject the null hypothesis; $n$, denominator for rate.

although the incidence rate appeared higher, but more cases had early-stage disease with SS− TB. Also, these results corroborate previous findings that multiple TB screenings in individuals with initial abnormal CXRs result in the detection and identification of more SS− TB cases at an early disease stage (*Kuan, 2018*), which might benefit for timely therapy initiation; thus, might increase the success of treatment and reduce disease burden including blocking the potential disease dissemination. However, the high proportion of bacteriological negativity that is, of >35% with B- TB in both groups, for example, 56.4% in labor migrants vs. 39.2% in marriage migrants (Table 1), has suggested the multiple TB screenings might be of over-detection (*Kuehne et al., 2018*; *Aldridge et al., 2016*). While the reasons for the high TB burden in the migrant population are likely to be the reactivation of remotely acquired LTBI following migration from high TB burden countries to lower TB burden countries (*Global Tuberculosis Report, 2018*; *The World Bank, 2018*; *Taiwan Centers for Disease Control, 2018*). Therefore, it is important that

applying a LTBI screening combined with prevention treatment (PT) at a pre- or post-entry in the very beginning will save a lot of efforts in later stages, for example, multiple screenings (*The New York Times Editorial Board, 2018*) and might meet the core of an earlier TB mitigation strategy for TB control intervention targeting at high-risk migrants even including BCG-vaccinated individuals (*Olivieri et al., 2016*); for example, a 2-step LTBI screening: performing an expensive interferon-gamma release assay (IGRA) after positivity on the economical tuberculin skin test (TST), then combined with PT (*Olivieri et al., 2016*). Thus, antibiotics can effectively and economically eliminate tuberculosis, before they become contagious, that is, to treat individuals who are still invisibly sick. In several countries, such as the United States, Britain, and Canada, LTBI screenings combined with PT strategies have long since become public health norms for migrants (*The New York Times Editorial Board, 2018*).

The repatriated labor migrants with TB were not traced in this study if they received any treatment when they returned home country. Nonetheless, both marriage migrants and labor migrants with TB who stayed in Taiwan and accepted either the standard (6–9 months) or longer regimen of TB therapy achieved a TB treatment completion rate of 87–99% (Table 1), which was above the WHO target of >85% (*WHO, 2006–2015*) during 2012–2015. In terms of assessing the overall treatment outcome of TB cases, it was found that cases identified among labor migrants significantly had poorer treatment completion than those among marriage-migrants in Taiwan because of a higher percentage of transferred out cases in 2012–2015 (Table 1). The treatment completion rate in Taiwan was high in marriage migrants (96–99%; Table 1). As for labor migrants, after the implementation of the therapy policy which eliminated repatriation of TB cases and allowed for therapy in the host country, the completion rates raised to 24–37% from 3.8–9.7% (Tables 1 and 3) after the implementation of the therapy policy during 2012–2015.

Overall, the improvement of an increased 4.88-fold that is, 30.9% in 2014–2015 vs. 6.7% in 2012–2013 of the TB treatment completion among labor migrants during periods post-implementation (2014–2015). It was further observed that TB cases with more SS− (OR: 7.45) or B− 44(OR: 1.48) and more SS+ (OR: 13.43) or B+ (OR: 5.83) was treatment completed since 2014 (Table 3) among labor-migrants, which demonstrates the benefit of implementing the friendly therapy policy. Also, the reducing potential structural barriers to TB treatment completion (*Kourbatova et al., 2006*; *Datiko & Lindtjorn, 2010*; *Lambert et al., 2003*), limited access to care (*Kourbatova et al., 2006*), and relocations of labor-migrants were conducted by introducing the therapy policy since 2014. Moreover, after 2014, a significant decline in transfer out TB cases was observed in both of SS- TB cases (OR: 0.61) or B− TB cases (OR: 0.58) and SS+TB cases (OR: 0.44) or B+ TB cases (OR: 0.66) among labor migrants during this study period (Table 3).

Additionally, because the higher number of SS− or B− TB cases among labor migrants than among marriage migrants was also worried as a potential risk of being delayed or untreated and then, in turn, developing TB dissemination. Since very few bacilli are sufficient to cause infection (*Sepkowitz, 1996*; *Scandurra et al., 2020*), therapy should not be delayed in labor-migrants with SS− TB according to the WHO guidelines, especially those

who were to provide long-term care for vulnerable people or the elderly; therefore, the initiation rate of anti-TB treatment within 30 days of diagnosis was concerned. After introducing the friendly therapy policy, the overall treatment initiation rate within 30 days of diagnosis was significantly increasing (OR: 1.31) among labor migrants. Furthermore, the overall initiation rate of treatment exhibited higher among labor migrants with SS+ TB (93.0%) or B+ TB (83.9%) than those with SS− TB (70.9%) or B− TB (77.4%) and implicated that relatively more TB cases with SS− or B− had somehow delayed treatment than those with SS+ or B+. Nevertheless, after introducing the therapy policy for labor migrants, showed a significant increase in the initiation rate of treatment both for SS− TB of 77% vs. 71% (OR: 1.38) or B− TB of 78% vs. 77% (OR:1.64) during the post vs. pre-implementation of the policy. Relatively, this implementation was not significantly impacted the treatment initiation rate among labor immigrants with SS+ TB of between 91–89% (OR: 0.63) or B+ (OR: 0.84) (Table 3). Thus, the friendly therapy policy in Taiwan has resulted in an improved experience of reducing structural barriers to TB mitigation since its implementation in 2014, which has successfully promoted the anti-TB treatment outcomes including improvement in treatment initiation especially those who with S− TB or B− TB and increasing treatment completion for those migrants with TB stayed in Taiwan. Therefore, based on our observations, there is a need for intensifying health education that promotes TB therapy includes delivering the information of the ongoing availability of free, accessible health services for vulnerable groups such as high-risk migrants; this health education could also be a critical element in increasing treatment success (Hayward et al., 2018; Scandurra et al., 2020; Dangisso, Datiko & Lindtjørn, 2015) and early therapy for individuals with TB in receiving countries.

## Limitations

The post-entry screening frequency of 1–2 rounds for marriage migrants was not the same as that for labor migrants (four rounds), which might cause an underestimation of TB incidence in the former. The authentic treatment completion rate may have been underestimated among labor migrants who opted to return to their original countries for treatment, as they were not or enrolled in or followed by this study. The proportions of migrants with TB who were due to TB reactivation or transmission were not defined by the molecular testing in this study.

## CONCLUSIONS

Multiple screenings following an initial abnormal CXR in migrants could detect early-stage TB cases; nevertheless, an improved therapy completion is a substantial step for TB elimination. The relatively higher odds of SS+ TB and bacterial negativity >35% among migrants might be an index of persistent TB reactivation or over-diagnosis; therefore, it is recommended that adding LTBI screening combined with preventive treatment as an alternative approach might save multiple screening effort for high-risk migrants. The friendly treatment policy for migrants, which eliminated repatriation for labor migrants with TB, could benefit in anti-TB treatment outcomes include increasing therapy

initiation in SS− or B− TB cases and treatment completion for those who stayed in receiving countries.

## ACKNOWLEDGEMENTS

We thank all Taiwanese clinics for their routine data notification and the Taiwan CDC staff for disease control administration and data management. We would like to express deep gratitude and appreciation to Prof. Mesay Hailu Dangisso and Dr. Dustin C. Yang for reviewing the manuscript and offering expertise suggestions.

### Funding

The publication fee would be paid by Taiwan CDC. The funders had no role in study design, data collection and analysis, decision to publish, or preparation of the manuscript.

### Grant Disclosures

The following grant information was disclosed by the authors:
Taiwan CDC.

### Competing Interests

The author declares that they have no competing interests.

### Author Contributions

- Mei-Mei Kuan conceived and designed the experiments, performed the experiments, analyzed the data, prepared figures and/or tables, authored or reviewed drafts of the paper, and approved the final draft.

### Ethics

The following information was supplied relating to ethical approvals (i.e., approving body and any reference numbers):

This study was approved by the Institutional Review Board of the Taiwanese CDC (No. TwCDCIRB106115).

### Data Availability

The raw data summary is available as a Supplemental File.

The raw data used in this study were acquired from the TCDC TB registry system which cannot be shared due to the individual information privacy protection policy. Interested readers can apply for access to the data resources: details of individual TB cases including "area, age, and gender".

Interested readers can access the TCDC Epidemic Intelligence Center data resources by contacting gnnhuo@cdc.gov.tw to obtain access to the data presented here: https://data.cdc.gov.tw/en/dataset/aagstable-tuberculosis (and graphed here: https://nidss.cdc.gov.tw/en/nndss/Diagram?id=010).

## Supplemental Information

Supplemental information for this article can be found online at http://dx.doi.org/10.7717/peerj.10332#supplemental-information.

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
