# Peer review of "Surveillance of tuberculosis and treatment outcomes following screening and therapy interventions among marriage-migrants and labor-migrants from high TB endemic countries in Taiwan"

_PeerJ, doi:10.7717/peerj.10332_

## Round 0.1 · original submission · Major Revisions

The authors should revise carefully the manuscript paying particular attention to the suggestions related to the statistical analyses.

Reviewer 1 ·

Basic reporting

This statistical analyses and results presentation remain not reach the basic standard for a scientific manuscript, so I am not able to verify the validity of this study. I highly recommend the author to find a statistical consultant and go through the analytic plans and the result tables carefully.

Experimental design

NA

Validity of the findings

See basic report

·

Basic reporting

Dear Academic editor,

Thank you for inviting me to review this manuscript “Surveillance of tuberculosis and treatment outcomes following screening and therapy interventions among marriage migrants and labor migrants from high TB endemic countries in Taiwan (#47696)”.

I have comments and questions for the authors to improve their manuscript.

Review comments

Major comments
1. The justification for comparing only treatment outcomes of marriage migrants and labour migrants from TB high endemic countries should explained. Why the authors did not compare the treatment outcome with Taiwanese population data?
2. The study did not use the same assessment criteria to assess TB trends. They used 1-2 rounds of post-entry TB screening for marriage migrants and 4 rounds of Post-entry TB screening. The authors should justify this.
Minor comments
Abstract
The authors used labor migrant as different comparison groups, how do the authors know marital status of labour migrants?
Is this post-entry ?
… “with 61% bacteria-positive and 31% smear-positive cases..” what is the difference between smear-positive and bacteria-positive? Smear-positives are also bacteria positive even if they are culture positive.

“TB cure/treatment completion” use either treatment completion or cure rate. Cure rate and completion are different.

In statistics and analysis section

…..“The TB incidence (per 100,000 population) was calculated as the number of newly detected 114 labor migrants with TB during the study period (2012-2015) divided by the population per year”…...
Is the population (denominator) the total population of Taiwan or total labour migrant population? This needs to be clear for the readers

Line # 125-126. The authors estimated the treatment outcomes by considering transferred out cases and treatment completion rate only. For international comparison why the authors did not estimated the lost-to-follow up cases?
Lines # 151, 168, . postentry…is this post-entry? if so ,correct this throughout the manuscript

Why the authors presented both Pearson’s correlation and linear association?

Why the authors did not categorize TB cases as bacteriological confirmed and smear-or bacteriologically negative cases?

Discussion

General comments; no need of presenting the results here in the discussion. All the statistics were already presented in the results section

Line # 246. ….after 2014 vs. before 2013 being less transfer out.. isn’t this less transferred out? not clear

The authors used only ss- and ss+ cases in the discussion of treatment outcomes. What about bacteriologically confirmed cases; culture positive cases. The authors could consider merging either ss+ and culture positive cases together or include bacteriologically confirmed cases in the discussion.

Limitation
“The proportions of migrants with TB who had reactivation or transmission cases were not defined by the lab”. This statement is confusing and not clear.

Table 1. In clinical characteristics, the authors analysed and reported B+ and SS+ separately. The authors need to make clear whether SS+ cases were included in the B+ cases.
Table 2. Under CXR; “Abnormal w/o cavitation” and “Abnormal w/ cavitation, should be written in full.

Under sputum culture. CC+, and CC- should be made clear for the readers new for the field

Experimental design

Original primary research within Aims and Scope of the journal.
Research question well defined, relevant & meaningful. It is stated how research fills an identified knowledge gap.
Yes

The submission should clearly define the research question, which must be relevant and meaningful. The knowledge gap being investigated should be identified, and statements should be made as to how the study contributes to filling that gap.
Rigorous investigation performed to a high technical & ethical standard.
Yes

The investigation must have been conducted rigorously and to a high technical standard. The research must have been conducted in conformity with the prevailing ethical standards in the field.
Methods described with sufficient detail & information to replicate.

Yes, with minor revision

Methods should be described with sufficient information to be reproducible by another investigator.

Validity of the findings

Conclusions are well stated, linked to original research question & limited to supporting results.

With modification

---

## Round 0.2 · accepted · Accept

The revision has been properly performed in the manuscript, but next time, please check carefully the response to reviewers you wrote.